# Comprehensive Analysis of Hydrological Processes in a Programmable Environment: The Watershed Modeling Framework

Nicolás Velásquez [1,*], Jaime Ignacio Vélez [2], Oscar D. Álvarez-Villa [3] and Sandra Patricia Salamanca [2]

[1] Department of Civil and Environmental Engineering, Iowa Flood Center, University of Iowa, Iowa City, IA 52245, USA
[2] Department of Geosciences and Environment, Universidad Nacional de Colombia, Sede Medellín 050034, Colombia
[3] Emergente Sustainable Energy, Envigado 055422, Colombia
[*] Correspondence: nicolas-giron@uiowa.edu; Tel.: +1-319-512-4194

**Abstract:** Distributed hydrological modeling has increased its popularity in the community, leading to the development of multiple models with different approaches. However, the rapid growth has also opened a gap between models, interfaces, and advanced users. User interfaces help to set up and pre-process steps. Nevertheless, they also limit the implementation of more complex experiments. This work presents the Watershed Modeling Framework (WMF) as a step forward in closing the interface–usage gap. WMF is a Fortran-Python module designed to provide tools to perform hydrological analysis and modeling that conceptualizes the watershed as an object with a defined topology, properties, and functions. WMF has a built-in hydrological model, geomorphological analysis functions, and a QGIS plugin. WMF interacts with other popular Python modules, making it dynamic and expandible. In this work, we describe the structure of WMF and its capabilities. We also provide some examples of its implementation and discuss its future development.

**Keywords:** hydrological modeling; Python; Fortran; streamflow topology; watershed analysis

## 1. Introduction

In recent years, we have experienced a significant increase in the data, software, and hardware available to perform hydrological modeling and analysis. At present, depending on the region of the world, Digital Elevation Model (DEM) products with high resolution are available, such as ALOS-PALSAR [1] and NHDplus [2]. In addition, there are public processed streamflow networks such as Hydrosheds and HYDRO1K [3,4], soil data, and land use data such as MODIS, LANDSAT, and SENTINEL [5,6]. Furthermore, there has been a significant increase in time series derived from remote sensor sources such as GRACE, IMERG, APHRO, and MSWEP. Additionally, the quantity of computational resources and available software products have also significantly increased. Considering the available resources, it has become common for hydrologists to perform more complex analyses involving larger amounts of data. The described data increase highlights the need for software that provides straightforward communication and the bases for replicable and scalable experiments that can be customizable.

On the software side, plenty of open and commercial packages offer tools to perform hydrological analysis and modeling. GIS software such as GRASS [7], SAGA-GIS [8,9], and QGIS [10] are some of the most popular free software products that provide tools to perform hydrological analysis. Some examples include the TOPMODEL incorporated in GRASS [11,12] and the inclusion of SWAT [13] in ArcGIS. Other hydrological models, such as VIC [14,15], HBV [16], and HEC-HMS [17], offer stand-alone and plugin-like options for their execution. The mentioned applications are robust and well-known by the

community. However, in many cases, their use is limited by their interface (if any) and the input and output formats. Different efforts have been made to tackle input–output communication through communication frameworks [18–20]. Some developments also exist to bring together the models' building blocks [21–23]. However, limitations still make implementing complex analysis, automation, and replicability difficult. In addition, models usually lack an open interface that works on a high-level programming language such as Python.

We developed the Watershed Modeling Framework (WMF) package to address some of the challenges mentioned above. We used Fortran 90 to perform the intensive numerical tasks and Python 3.7 for the interface. We connected both languages using f2py [24]. WMF is an open tool that facilitates hydrological analysis and simulation using a high-level programming language. The package includes tools to extract watersheds, analyze them, and simulate several hydrological processes. WMF uses arrays to represent and store the topological connection of the cells and hillslopes that belong to a watershed. Based on both topologies, WMF offers functions to perform hydrological simulations and compute watershed descriptors such as HAND [25], rDUNE [26], and the topographic index [27]. Additionally, it has functions to interact with spatiotemporal data, including point-measured data (rainfall stations, level gauges, etc.) and remotely sensed data (rainfall, soil moisture).

Moreover, WMF can read and write data in several GIS raster and vector formats. We developed WMF to interact with pandas DataFrames [28] and NumPy arrays [29]. Since WMF works directly on Python, it is editable and can be coupled with external software packages allowing the user to build on top of the framework. The described characteristics make WMF a powerful tool that can expand with the users' developments while contributing to the hydrological community. In addition to the Python interface, WMF has a QGIS plugin that works as a GUI (Graphical User Interface).

This manuscript first presents the design and structure of WMF, including its general architecture and the details of its main modules and elements. Then, it describes the QGIS plugin. The Section 3 presents examples and discusses several results obtained with WMF. Finally, we discuss the tool's current limitations and future development.

## 2. Materials and Methods

We wrote the core modules of WMF in Fortran 90 and the interface to it in Python 3 (Figure 1). The core modules *cuencas.f90* and *models.f90* perform the computationally intensive algorithms while the Python script acts as a user interface. The library *cuencas.f90* includes 70 functions to perform tasks involving streamflow delineation, watershed delineation, and analysis (Figure 1a). In addition, *models.f90* contains the distributed hydrological model and three sub-models to simulate erosion, shallow landslides, and flash flood processes (Figure 1b). The Python interface (*wmf.py*) acts as a link to the Fortran modules. During its installation, the Fortran code is compiled and warped to interact with Python using f2py [24]. Additionally, using Python, we developed the QGIS plugin or WMF-Q (see Section 2.4). In the following, we describe some of the built-in functions and models.

### 2.1. Watershed Extraction and Topology

In WMF, we conceptualized the watershed as a Python class with several characteristics and methods. The most relevant features are the cells' and hills' topological connectivity ($C_t$ and $H_t$, respectively). WMF uses a DEM and its corresponding direction map DIR (see Figure 2) to extract $C_t$ for a given output. The DEM and DIR maps must have a format supported by GDAL [24]. The DIR downstream connectivity must follow the numbering presented in Figure 2. However, each DEM processor provides a different connectivity convention, such as TauDEM [30] or the AT algorithm [31]. To address this issue, we included a function to translate DIR maps from different sources to the format required by WMF. After extracting $C_t$, WMF uses an upstream area threshold to delineate the hillslopes and their connectivity $H_t$.

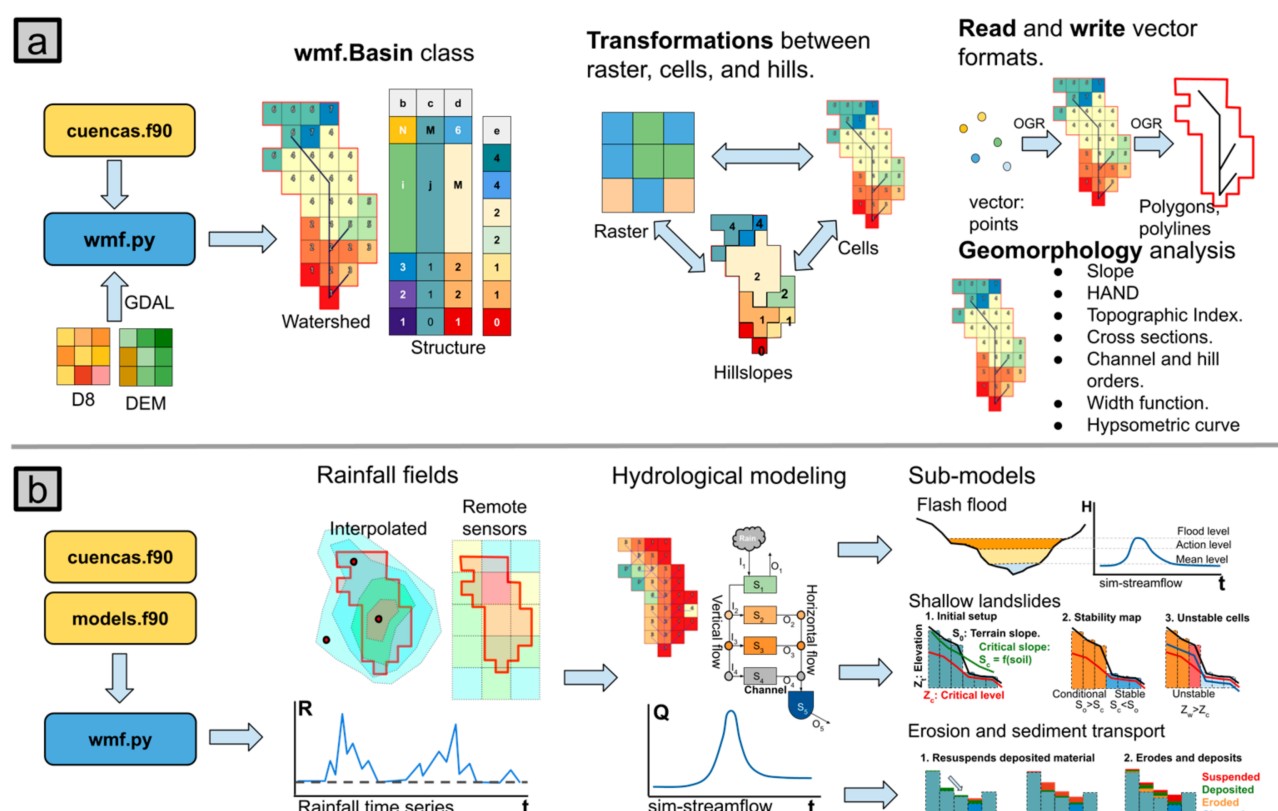

**Figure 1.** The general structure of WMF. wmf.py contains the functions to interact with the Fortran modules: (**a**) presents the interaction with the module *cuencas.f90*, which allows watersheds to be defined; (**b**) describes the interaction with the *models.f90* and its tools to perform hydrological modeling.

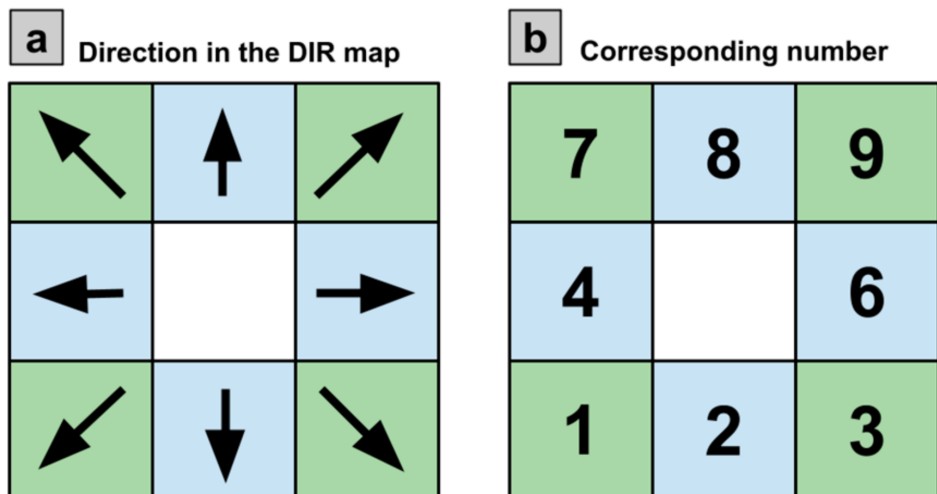

**Figure 2.** Flow direction map (DIR) numbering conventions for WMF. The panels show the flow's physical direction (**a**) and the corresponding number (**b**). Blue and green colors represent orthogonal and diagonal flows, respectively.

WMF uses a recursive algorithm to determine the elements that belong to $C_t$ using the DIR connectivity (Figure 3a). The search starts in the outlet cell by finding the neighbors draining towards it (the cell marked in red in Figure 3b). In this case, according to Figure 3a, the parent cells of cell 1 are 2, 3, and 4. The parent cells are added to the list of cells that belong to the watershed. Then the algorithm moves to the next cell (2 in this case), repeating the search performed in the outlet. The list does not increase if the cell has no parents (the

case of cell 2). The algorithm finishes after searching for the parents of all the elements added to the list. At this point, WMF has the list of cells that belong to the watershed and their downstream neighbor. The list is organized so that the most upstream elements are located at the beginning of the list (Figure 3f). Figure 3c describes the downstream neighbor obtained at the watershed extraction's end.

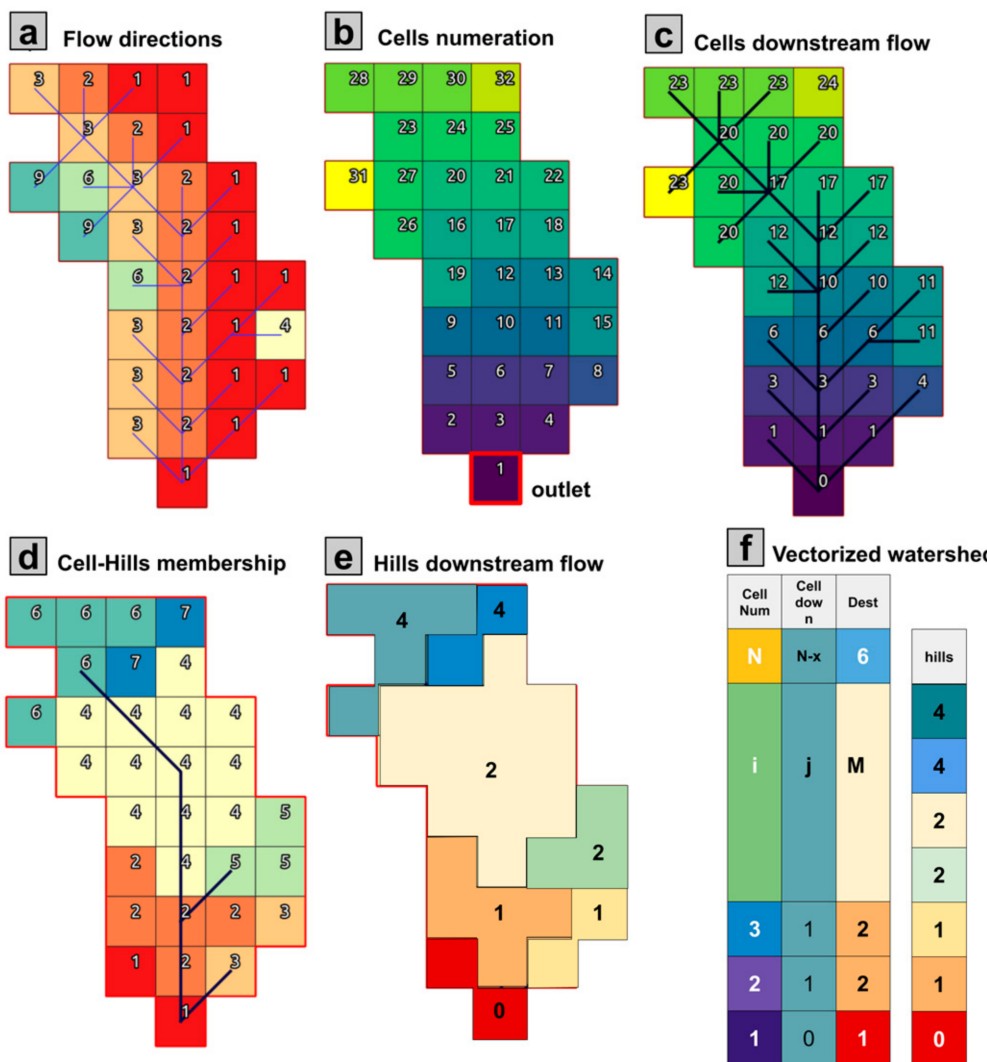

**Figure 3.** The base topological structure of a watershed in WMF. (**a**) Initial DIR map, blue lines denote directions, the colors represent the direction; (**b**) watershed cell numeration relative to the outlet cell, colors represent the cell number. (**c**) Cells downstream flow relative to the cell numeration, colors represent the cell number. (**d**) Cells membership to identified hills based on threshold area to determine network, colors represent each hill. (**e**) Hills downstream flow relative to the outlet hill, colors represent each hill. (**f**) Scheme of the vectorized properties of the watershed shown from panels (**b**) to (**e**).

The hillslope connectivity $H_t$ is obtained from the cells' connectivity array. Using $C_t$, WMF computes the upstream number of accumulated cells and, depending on a threshold value, determines which are part of the network (black lines in Figure 3d). Then, the cells that drain to each network segment are assigned to each hillslope element (number in Figure 3d). Finally, the hillslope connectivity is inherited from the network (Figure 3e). Figure 3f summarizes the resulting products in an array shape (their representation in WMF). In the figure, columns b, c, d, and e correspond to the cell numbering, downstream connectivity, cells hillslope belonging, and hillslope connectivity, respectively. The de-

scribed array structure is the base of multiple WMF operations, such as geomorphological analysis, hydrological modeling, and GIS transformations.

The array type definition of the watershed topologies ($C_t$ and $H_t$) is one of the most relevant outcomes. Regardless of the predefined functions, $C_t$ and $H_t$ allow easy implementation of new functions inside Python, Fortran, or other languages. When executed, the cell connection topology $C_t$ is defined in WMF as *basin.structure*, and $H_t$ as *basin.hills*. The connection between $C_t$ and $H_t$ is stored in *basin.hillsown*. Additionally, the *basin* class holds the georeferencing properties allowing WMF to interact with maps.

### 2.2. Watershed Functions

The basin class functions allow interaction with raster and vector maps, performing geomorphological analysis, and executing a distributed hydrological model along with three sub-models. In the following, we summarize the main functions included in the class.

### 2.2.1. Data Interaction

Once the watershed is defined, WMF counts with functions to transform and read vector and raster data allowing their interaction with external data. The *Transform* functions collection allows WMF to interchange information between raster maps and the *basin* class. It also provides an interface to convert values between hillslopes and cells. Table 1 summarizes and describes the available transformation functions.

**Table 1.** Basin class transformation functions.

| Function Name | Description | Result |
|---|---|---|
| Transform_Map2Basin | Converts raster data and resamples it to the cells of the basin. | Basin.array |
| Transform_Basin2Map | Converts any basin variable to a map and writes it to the disk. | Gdal raster |
| Transform_Hills2Basin | Maps a hillslope variable to their corresponding cells. | Basin.array |
| Transform_Basin2Polygon | Saves a discrete cell variable into a collection of polygons. | Polygon |
| Transform_Basin2Hills | Maps a cell's variable to their corresponding hillslopes. | Hills.array |

In addition to the transformations, the basin class has functions to save results in a vector data format. The function *Save_Net2Map* allows saving the network into a vector layer in which each channel segment has its link ID, stream order, and length. Moreover, the function can save additional features using a dictionary containing their names and an array with values. Additionally, the function *Save_Basin2Map* allows writing a vector layer with the watershed boundary.

### 2.2.2. Geomorphology

WMF has a set of functions to perform automated geomorphological analysis. Almost all functions work over the cells array; however, a representative portion of the results can be transformed into hills or links using the cell-hills membership element (Figure 3d) and NumPy functionalities. Table 2 describes the collection of functions to perform basic geomorphological analysis. The functions use the *basin* class as the main argument and operate over the *basin.structure* array described in Section 2.1. The functions store the results as a property of the *basin* class under its name. As an example, *basin.GetGeo_Parameters()* obtains the variables *basin.GeoParameters* and *basin.Travel_time*. Most of the functions described here use the Fortran module *cuencas.f90.* However, functions can also be written in Python using the NumPy package and the attribute *basin.structure*.

Additionally, WMF has a set of more advanced functions (Table 3). These allow for estimating isochrones, the width function, and the hypsometric curve. It also has functions to estimate distributed variables, such as the topographic wetness index [27], the HAND model [25], and the rDUNE model [26]. Most functions use only the *basin.structure* as their main argument. However, some of them require additional information provided by the user. In addition, compared with the basic collection, some of the advanced functions may take more execution time.

**Table 2.** Basin class basic geomorphological analysis function collection.

| Function Name | Description | Results |
|---|---|---|
| GetGeo_Parameters | Computes a collection of geomorphological parameters such as the watershed area, its mean slope, main channel length, hypsometric curve, stream density, and concentration time. | GeoParameters: area, slope, centroid, length. Travel_time: Mean travel time computed using different equations. |
| GetGeo_Cell_Basics | Computes each watershed cell's upstream area, length, slope, and height. | CellAcum: Upstream cell accumulation [-]. CellLong: Length of each cell [m]. CellSlope: Slope of each cell [m/m]. CellHeight: Elevation of each cell [m]. |
| GetGeo_StreamOrder | Computes the Horton stream order of each network element. | CellHorton_Hill: Horton order of each hillslope. CellHorton_Stream: Horton order of each stream. |

**Table 3.** Basin class advanced geomorphological analysis function collection.

| Function Name | Description | Results |
|---|---|---|
| GetGeo_IsoChrones | Computes an approximation of the travel time of each cell using as an argument the travel time estimated for the watershed. | CellTravelTime: the travel time of each cell [h]. |
| GetGeo_WidthFunction | Computes the width function of the watershed as the distance between each element and the outlet. | Width_distance: distance between each network element and the outlet. |
| GetGeo_Ppal_hipsometric | Computes the hypsometric curve along the main channel of the watershed and among all the watershed elements. | Hipso_ppal: Hipsometric curve along the main channel. |
| GetGeo_IT | Computes the topographic index (IT) for each cell of the watershed. | Hipso_basin: Hipsometric curve using all the cells. |
| GetGeo_HAND_and_ rDUNE | Computes HAND and rDUNE values for each cell. | IT: the topographic index |

### 2.3. Hydrological Modeling

WMF also has tools to perform distributed hydrological modeling. The module *models.f90* contains a distributed hydrological model and three sub-models (Figure 4b). Additionally, it has functions for pre-processing and post-processing results.

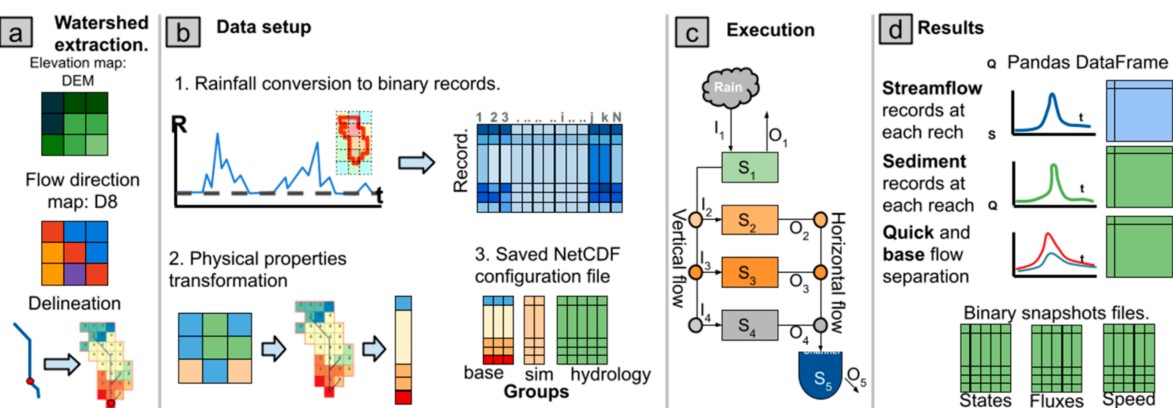

**Figure 4.** WMF hydrological model architecture. (**a**) Watershed extraction from a DEM and a DIR map. (**b**) Data pre-processing to obtain binary rainfall data (top blue matrix) and the netCDF file that stores the watershed topology and the model configuration. (**c**) Hydrological model execution. (**d**) Results are obtained in pandas DataFrames for time series and binary files for fluxes, states, and speed.

In the current development, WMF has a hydrological model based on the distributed model TETIS [32,33]. The version at WMF has been written from scratch and has considerable differences compared to the original model. In the TETIS model, only the streamflow

speed is approximated with a kinematic wave equation. In contrast, WMF Hydro (WMF-H) uses this approach in every horizontal flow. It can also track rainfall inputs and separate surface and subsurface streamflow contributions. Moreover, runoff production could occur as a Hortonian process (intensity greater than infiltration) or by saturation. Finally, the current version accepts evapotranspiration as a forcing to the model.

The model represents the water dynamics by using five storage tanks. The tanks correspond to capillary storage (tank 1), surface storage (tank 2), subsurface storage (tank 3), aquifer storage (tank 4), and water in the channels (tank 5). Tanks 1 to 4 are present in all the elements, and tank 5 presence depends on the existence of a channel. The rain falls over tank 1 and flows to tanks 2, 3, and 4. Moreover, depending on the cell type (hill, rill, or channel), a portion of the water from tanks 2 and 3 will go to the same tanks at the downstream cell or tank 5 (channel) of the same cell. In the case of tank 4, the water gradually goes to tank 5. The vertical connections of the tanks are represented as linear processes, whereas horizontal processes could be linear or non-linear. For a further description of the model, please read [32–34].

Additionally, WMF has an architecture to manage the model's pre-processing and post-processing data (Figure 4). Pre-processing starts with the extraction of $C_t$ and $H_t$ from the DEM and the DIR maps (Figure 4a). With $C_t$ and $H_t$ defined, WMF can pre-process the forcing data and assign parameter maps (Figure 4b). The forcing data are stored in a binary file and a header file. The binary file is divided by records, each corresponding to a time step with an input greater than zero. The header file indicates the date and mean rainfall value of each record. After the pre-processing, WMF allows the execution of the hydrological model (Figure 4c). The user can customize the execution when calling *basin.run_model* and change global model variables by directly editing them. Finally, the execution generates a report and a pandas *DataFrame* with the simulated flows of each river segment. Depending on the configuration, the WMF can provide additional outputs (Figure 4d). Additionally, some setups allow the model to write binary files with time-variable snapshots of the model states and fluxes (bottom of Figure 4d) and the results of the sub-models.

### 2.3.1. Shallow Landslides Sub-Model

We adopted the shallow landslide sub-model proposed by [35]. Each cell's stability is calculated by assessing the different stresses applied to the soil matrix in the landslide sub-model. The landslide sub-model estimates the soil's stability in the function of the pore water pressure [36], which relies on the soil water content. At each time step of the execution, WMF-H computes the water content at tank 3 (subsurface), which is then used by the landslide sub-model to estimate the stability of each grid cell.

The landslides model requires additional parameters related to the soil properties. From these parameters, WMF obtains a stability map that classifies cells into conditionally stable, unstable, and stable. Then, during execution, the model only evaluates the conditionally stable cells. Ref. [35] gives a more detailed description of the model, and [34] presents an example of its implementation in WMF during a flash flood case in Salgar (Antioquia, Colombia) in 2015.

### 2.3.2. Erosion Sub-Models

WMF offers two sub-models to simulate sediment erosion and transport processes: steady-state and dynamic models. The steady state model is the RUSLE (Revised Universal Soil Loss Equation) [37], while the dynamic model corresponds to the CASC2D-SED sediment component [38,39]. The RUSLE uses parameters and topographical factors, while the dynamic erosion model relies on the hydrological model. There is a significant difference in the level of complexity of both models. However, despite both approaches' conceptual differences, ref. [40] showed that their results are comparable in the long term.

The RUSLE and its derivations are the most frequently used models for soil erosion estimation. The spatial analysis tool allows computing the RUSLE using loaded maps of soil properties and maps derived from the landscape, for which the LS (Slope Length and

Steepness) is one of the most relevant. Considering this, WMF allows the user to compute LS using five different techniques since the RUSLE is sensitive to the LS parameter [37].

By comparison, a dynamic model is a conceptual approach to the erosion and transport of sediments. The erosion process happens only in the hills, while transport occurs in the hills and channels. The model assumes three particle sizes: sand, lime, and clay. The model first computes the total transport energy to determine the erosion using the runoff (tank 2). Then, it spends the transport energy moving suspended particles, depositing particles, and eroding. Once eroded, a particle becomes part of the suspended load, is transported downhill, or is settled as deposited storage. Ref. [38] describes the sediment model in detail.

### 2.3.3. HydroFlash: Flash Flood Sub-Model

We designed the HydroFlash sub-model to estimate floodplain inundations using the model outputs and the DEM definition. The module uses cross-sections along the river network, flow magnitude, and velocity. HydroFlash estimates the flooded cells of each cross-section at each time step by matching the hydraulic and hydrologic flow. The module also has a post-process filling algorithm applied at the end of each time step. See [34] for a more detailed description of HydroFlash and its implementation.

### 2.4. Q-Gis Plugin

WMF-Q is a QGIS plugin designed to use a GUI (Graphical User Interface) for WMF. At the current stage of development, WMF-Q runs on Linux and Mac machines. We developed WMF-Q using the PyQGIS Developer Cookbook. For the operation of WMF-Q, WMF must be installed in the same directory as QGIS. All files required for plugin operation are available on the *develop* repository of WMF. For its installation, the folder *qgisplugin* must be in the Python plugins folder of QGIS. Finally, the user activates the plugin from the QGIS plugin manager.

WMF-Q keeps the idea of working in a watershed as an element that has many properties and functionalities. At its start, the plugin allows users to open a previous watershed project saved in the netCDF format. It also enables loading the required elevation and direction maps to extract new watersheds. The plugin has five tabs: Configuration, Manager, Geomorphology, Hydrology, and Simulation. It also has the project area at the top of the tabs. Figure 5 presents the scheme of the described structure. The user can load previously saved watersheds and their computed variables in the project area. The Manager tab allows converting raster maps loaded in QGIS to the watershed structure. It also allows saving new variables into the netCDF file and includes a calculator to include Pythonic expressions. The Geomorphology tab has several predefined functions to compute the variables described in Section 2.2. The Hydrology tab has the tools to interpolate rainfall fields for a given watershed, transform radar data to match the watershed, estimate the water budget, and perform regionalization of extreme values. Finally, the Simulation tab offers the tools to set up the hydrological model and the erosion sub-models. It also contains tools to run the model and visualize the obtained results.

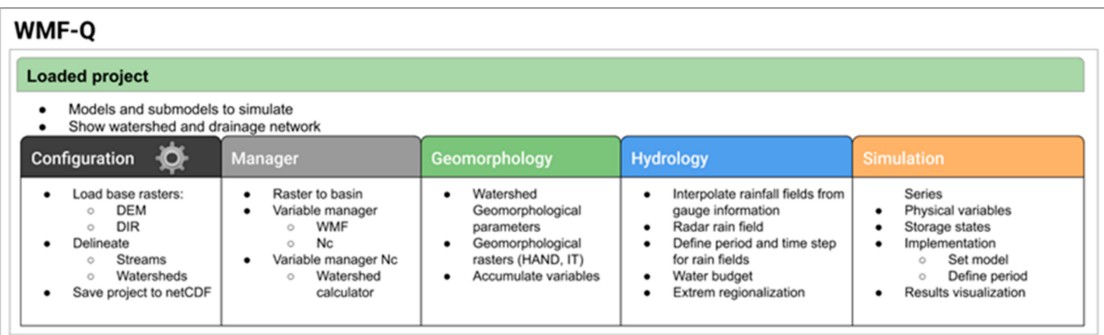

**Figure 5.** Description of the WMF-Q structure and its components.

## 3. Results and Discussion

This section presents examples of WMF and its usage for hydrological analysis and modeling. In the examples, we show the results derived from using some functions. Detailed use of WMF can be found in the following GitHub repository: https://github.com/nicolas998/WMF-Examples accessed on 2 March 2023.

### 3.1. Geomorphological Analysis

We used the Nishnabotna watershed (Iowa, USA) and a DEM with a pixel size of about 93 m for the geomorphological examples. We started reading the DEM and DIR maps in the analysis and delineating the watershed at the outlet point (−95.63, 40.64). Then, we computed the basic geomorphological features of the watershed using *GetGeo_Parameters()* and *GetGeo_CellBasics()*. From *GetGeo_Parameters()*, we obtained an estimated area of 7145 km$^2$, a perimeter of 325 km, a mean slope 3.9%, and a total channel length of 2475 km. The function also computes the travel time using the US Army Corps [41], Kiprich [42], Giandotti [43], Johnstone [44], Ventura [45], and Temez [46] equations, obtaining a median value of about 24 h. From *GetGeo_CellBasics()*, we obtained the accumulated area, the slope, the cell lengths, and the height.

#### 3.1.1. Hypsometric Curve and Width Function

Furthermore, we can compute the watershed's hypsometric curve and width function using *GetGeo_Ppal_Hipsometric()* and *GetGeo_WidthFunction()*. According to the results, the Nishnabotna watershed has an elevation gradient of about 200 m. Its hypsometric curve (Figure 6b) also corresponds to a watershed in development with a relatively high erosion potential. Finally, the width function (Figure 6c) shows that the watershed accumulates most of its area upstream, creating possible lags in its expected response time. We draw the given descriptions in Figure 6b,c. However, we require further analysis to determine the hydrological behavior of the Nishnabotna watershed.

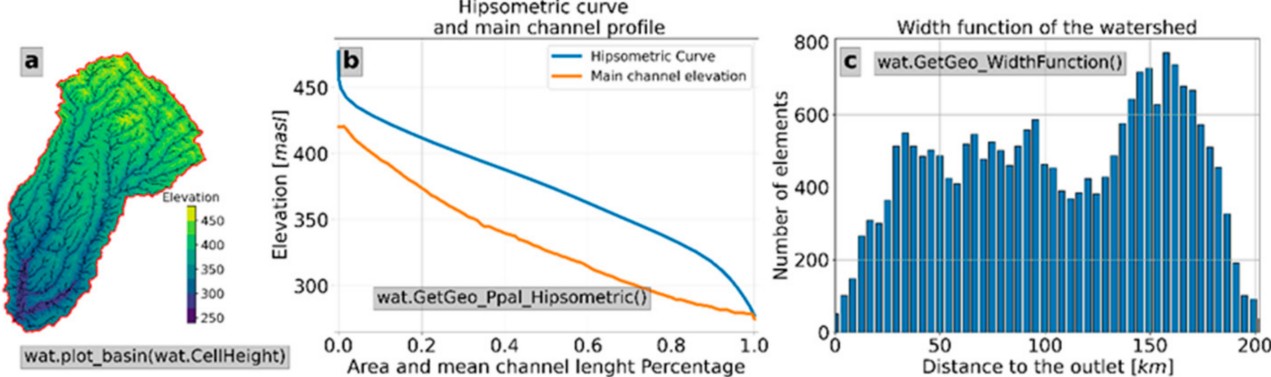

**Figure 6.** Example of some geomorphological properties obtained with WMF. Each panel presents a feature of the watershed and its corresponding function. Panel (**a**) shows the watershed, elevation, and streamflow network; panel (**b**) shows the hypsometric and the main channel profile, and panel (**c**) shows the width function.

#### 3.1.2. Distributed Geomorphological Analysis

The described results correspond to the fundamental geomorphological analysis of WMF. Other tools allow the computation of distributed parameters, such as the Horton order (Figure 7a), HAND (Figure 7b), the topographic index (Figure 7c), the hillslopes (Figure 7d), and the distance to the outlet (Figure 7e).

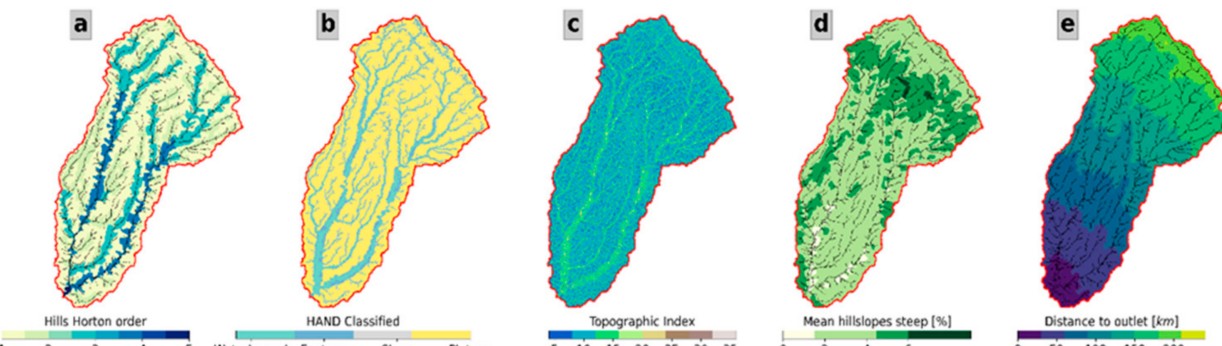

**Figure 7.** Example of geomorphological maps obtained using WMF. (**a**) Hills colored by the Horton order. (**b**) Cells colored using the [25] HAND classifications. (**c**) Topographic index for each cell of the watershed. (**d**) Hills colored using their mean hillslope. (**e**) Cells colored by the distance to the outlet.

According to Figure 7a, hill orders 1 and 2 cover most of the watershed area. Moreover, according to the HAND classification [25] (Figure 7b), orders 1 and 2 hillslopes are Plateau. We can also observe several areas classified as waterlogged near the outlet. The same area corresponds to areas that are prone to flooding in the watershed. The same regions exhibit a topographic index of around 15 (Figure 7c) and are more prone to accumulating water in the soils than the upstream regions. According to the slopes map (Figure 7d), hills are relatively flat except for the upper area of the watershed, where they exhibit values of around 6%. Finally, the distance to the outlet map (Figure 7e) presents a relatively uniform distance increment, reaching values up to 200 km. Following Figure 6c, the distance to the outlet map also suggests that a significant portion of the watershed develops upstream.

### 3.1.3. Advanced Analysis

In Figure 3d,e (Section 2.1), each described map is represented in WMF as an array allowing more advanced comparisons. Using WMF, we recreated the analysis presented by [26]. In this work, Loritz developed the reduced dissipation per unit length index (rDUNE) and compared it with HAND and the TWI. Figure 8 presents a similar task for the Nishnabotna watershed, discriminating by the Horton order (*wat.CellHorton_Hill*).

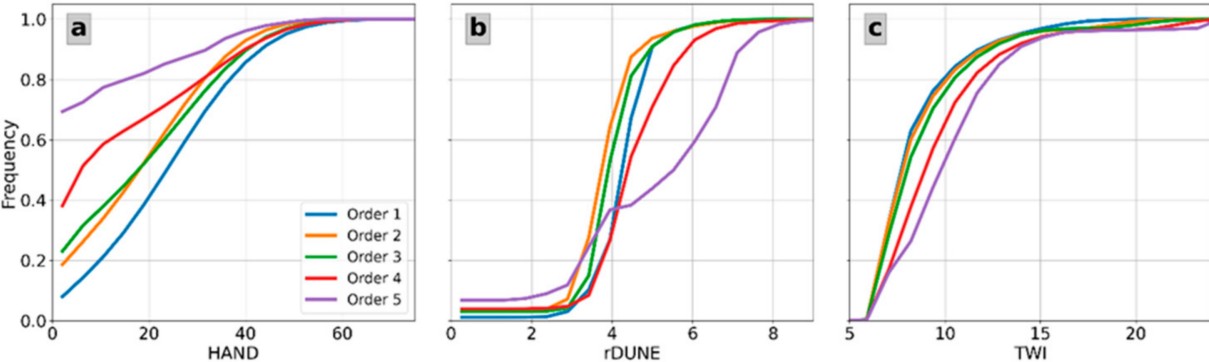

**Figure 8.** Cumulative distribution of the HAND (**a**), rDUNE (**b**), and TWI (**c**) discriminated by the Horton order of the Nishnabotna watershed.

According to Figure 8a, hill orders 5 and 4 have HAND values near zero and are more prone to be flooded. On the other hand, hill orders 3, 2, and 1 present HAND values above 30 m. The given description is helpful regarding floods. However, it provides a poor description of the degree of flashing of the hills. Conversely, rDUNE measures the potential energy dissipation. Accordingly, order 5 hills are the most dissipative, followed by order 4 hills (Figure 8b). It also shows that order 2 hills are the less dissipative hills, followed by

3 and 1. Finally, the topographic index describes where water is more likely to accumulate within the soil. In our case, the highest topographic indexes correspond to order 5 followed by orders 4, 3, 2, and 1 (Figure 8c). According to the described results, each index provides useful information regarding the watershed and subdivisions. In the example, we discretize by the Horton order obtaining significant differences across the cases. Nevertheless, users can configure the tool according to their needs.

The code and the data used for the development of this sub-section can be found in the examples repository: https://github.com/nicolas998/WMF-Examples/tree/main/Geomorphology_example, accessed on 2 March 2023.

### *3.2. Operations with Maps*

The following example uses the function *Transform_Map2Basin()* to compare the elevation and HAND properties with the temperature simulated by the High-Resolution Rapid Refresh (HRRR) [47] on August 1st at 1 A.M. Here, we first read a temperature raster map and its properties. We converted it to the watershed structure and finally compared it. After the transformation, WMF allows us to present the converted map within the watershed boundary (Figure 9a) and analyze it as an array (see the histogram in Figure 9b).

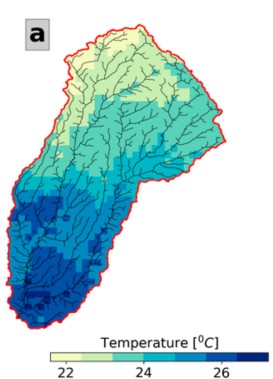
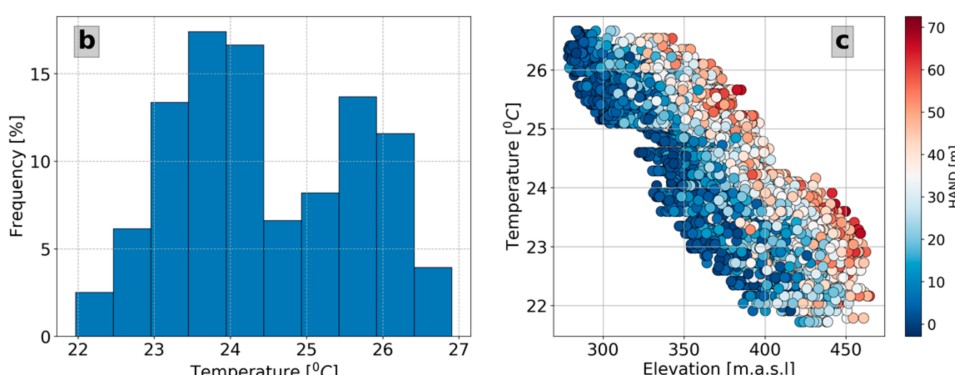

**Figure 9.** Analyzing temperature data in the watershed. (**a**) HRRR temperature spatial distribution along the watershed. (**b**) Histogram of the HRRR temperature. (**c**) Comparison between the elevation, temperature, and HAND.

The transformation also allows a direct comparison between the variable and the estimated properties of the watershed. To illustrate this, in Figure 9c, we compared the temperature map against the elevation and the HAND values (colors) in each cell. According to our results, temperatures drop with elevation. In addition, we also found that the HAND value partially explains temperature variability, a relationship likely related to the shading effect of the topography. Exploring further the described relationship is outside the scope of this work. However, the presented examples illustrate how WMF allows interaction with external data. The code and data of these examples can be found in the following GitHub repository: https://github.com/nicolas998/WMF-Examples/tree/main/maps_example, accessed on 2 March 2023.

### *3.3. Hydrological Model Simulations*

#### 3.3.1. Study Area and Modeling Goal

We used the utilities provided by the WMF to simulate the hydrologic dynamics in the Chinchiná river basin in Colombia. This modeling exercise was designed to estimate the potential of implementing reforestation-oriented Nature-based Solutions (NbS) in improving the basin's water regulation and its effect on the river discharges. The Chinchiná River basin is located on the western slope of the Central Cordillera of the Colombian Andes, South America (Figure 10A–C). The basin has an area of 1140 km$^2$ with altitudes ranging from 780 to 5400 m above mean sea level. It harbors nearly 550,000 people, most

of whom live in two cities (i.e., Manizales and Villamaría) and several small towns. The Intertropical Confluence Zone and the high-mountain topography are the main drivers controlling weather variability; these drivers generate slight fluctuations in the interannual temperatures, considerable changes in the intraday temperature, and a bimodal distribution of the annual precipitation. Temperature and precipitation oscillate between 12 and 18 °C and 1000 and 4000 mm per year. The basin also has several protected vulnerable ecosystems such as paramos, wetlands, and Andean forests [48].

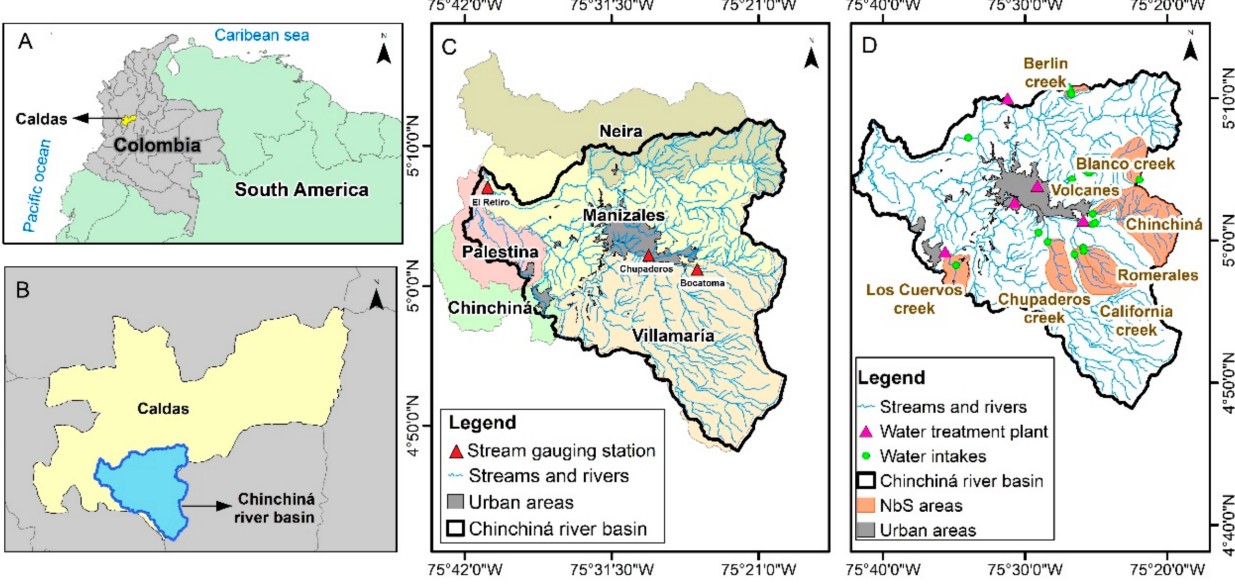

**Figure 10.** The geographic location of the Chinchiná river basin (maps (**A**) and(**B**)), the setting of cities and villages in the basin (map (**C**)), and areas where NbS will be implemented (map (**D**)).

The VivoCuenca Corporation (VCC) is the water fund operating for the Chinchiná River basin, whose mission is to collect and manage financial resources to preserve and restore the ecosystem services in the basin. VCC has developed three conservation portfolios to improve water regulation. These portfolios will operate mainly in the highly degraded sub-basins supplying the aqueducts (Figure 10, orange areas in map D). A combination of the following NbS forms each portfolio: (i) land acquisition for restoration, (ii) isolation of vegetation, (iii) revegetation, (iv) protection of the riparian buffer zone, (v) silvopastoral systems of different intensities, and (vi) agroforestry systems. The differences between the conservation portfolios are given by the selected NbS and their intensities of implementation. Portfolio 1 is the most intensive because it considers land acquisition, active restoration in large areas in the basin, and very intensive use of silvopastoral and agroforestry systems. Conversely, portfolio 3 is less intensive, considering protection and passive reforestation in riparian buffers, conserving previously existing protected areas, and implementing medium-intensive silvopastoral and agroforestry systems. Portfolio 2 consists of an intermediate solution between portfolios 1 and 3.

### 3.3.2. Scenarios Setup

Our modeling exercise was developed as follows. First, we gathered historical information to implement, calibrate, validate, and perform a sensitivity analysis for the TETIS hydrological model implemented in the WMF. After calibrating and validating the model, we set four scenarios for modeling future hydrologic dynamics from 2024 to 2054. The first scenario (trend scenario) considered the case in which the land use in the basin continues the historic degradation trend. Since the land cover is a dynamic variable, we predict the future evolution of land cover every five years using transition-probability-based cellular automata algorithms [49,50]. Therefore, all TETIS parameters related to

land cover are updated dynamically every five years in the simulation horizon. Figure 11 illustrates the effect of land cover change on the estimated maximum static storage from 2020 to 2052. Deforestation diminishes the maximum capacity in static storage in the trend scenario. Alternatively, taking the trend scenario as the reference, we directly represented the implementation of the three conservation portfolios in the trend land cover maps as new categories generated by vegetation growth. To establish vegetation growth rates, we consulted guides on natural vegetation in areas of Colombia with similar environmental conditions to the Chinchiná river basin [51].

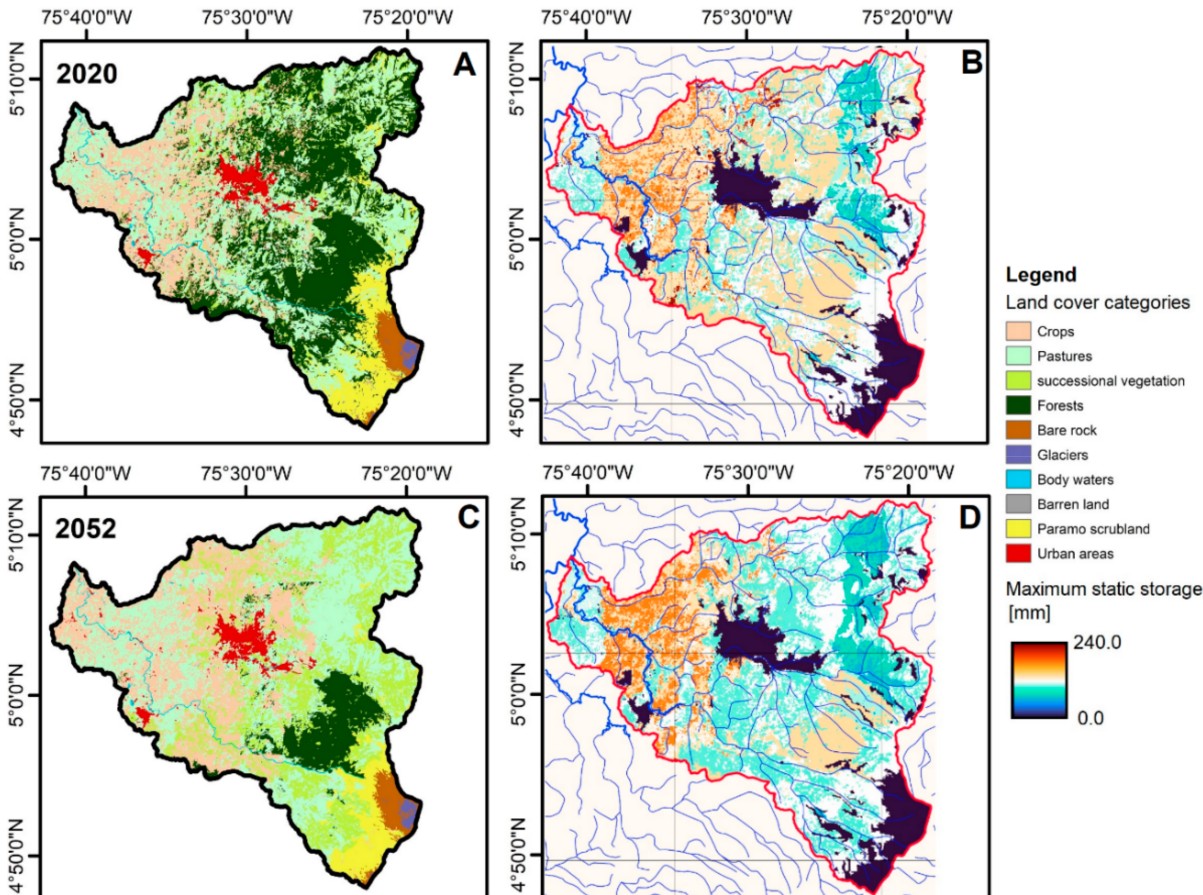

**Figure 11.** Schematical description of the land cover change in the Chinchiná River basin and its effect on a physical parameter of TETIS (maximum static storage). These results correspond to the trend scenario without implementing NbS conservation portfolios. Maps (**A,C**) present the land cover categories estimated for 2020 and 2052. Maps (**B,D**) show the respective estimated maximum static storage (*Hu*). Note that *Hu* decreases in many areas of the basin due to deforestation.

We compiled ground-based precipitation and temperature information from 115 rain gauges and 33 weather stations at a daily resolution with records from 1981 to 2022. We used densely sampled satellite databases such as CHIRPS (Climate Hazards Group InfraRed Precipitation with Station data) [52] and MODIS [53] to fill spatiotemporal gaps in the available data and improve the representativity of the hydrological processes. Similarly, we collected Landsat imagery to identify land cover categories in the Chinchiná river basin via random forest automatic classification. We could find few individual images completely lacking cloudiness in the basin, so we created ten composite images representative of selected years from 1989 to 2022. Historical land cover maps classified via random forest were used to train the predictive cellular automata models.

Since the main forcing of the model is precipitation, we performed a detailed characterization for the different modeling scenarios. Estimating historical daily precipitation

maps consists of three steps. First, we downscale CHIRPS precipitation from 1 km to 250 m pixel size using multivariate linear regression with bias correction [54]. Second, we estimate permissible spatial covariance models by merging inland rain gauge data with downscaled CHIRPS precipitation [55,56]. Third, we use kriging with an external drift [57] to estimate 250 m pixel size daily precipitation fields (2001–2022), which combine primary and secondary data and represent, as best as possible, the spatiotemporal variability of the precipitation in the basin. Likewise, for the NbS scenarios, we considered climate forcings (daily precipitation and temperature from 2024 to 2054) from the CMIP6 NORESM2-MM model [58], downscaled to 250 m pixel size by applying quantile correction [59]. Although many precipitation products from CMPI6 models do not accurately represent precipitation in tropical regions, [60] show that the NORESM2-MM model adequately represents the main features of the historical daily precipitation in Colombia.

The physical variables conditioning the hydrological modeling for the Chinchiná river basin are: (i) maximum static storage, (ii) maximum gravitational storage, (iii) reference potential evapotranspiration, (iv) infiltration capacity at a constant rate, (v) percolation capacity at a constant rate, and (vi) overland, interflow, in the channel, and base flow velocities. We created maps for all these variables in the basin. Maximum storage, infiltration capacity, and percolation capacity maps were estimated by employing a spatial superposition of historical land cover and soil texture maps [61,62]. Flow velocity fields were estimated using a general equation with the form $v_i = c_i A^{\alpha_i}$, where $i$ corresponds to the tanks, and A is the cross area of the tank. We used the Thornwaite equation expressed in a daily resolution to calculate the potential evapotranspiration maps. The input temperature to the Thornwaite equation was estimated via linear regression with altitude as the predictor variable.

To calibrate the model, we used the methodology presented by [33]. The selected objective functions are Nash–Sutcliffe (natural and logarithmic) and Kling–Gupta efficiency indexes. These indexes measure the model's performance on reproducing daily discharge series at the locations of three stream gauging stations: El Retiro, Bocatoma, and Chupaderos (see Figure 11C). The optimization procedure operates by testing values of ten corrector factors instead of directly modifying each tank's physical parameters. We used the Shuffle Complex Evolution (SCE-UA) algorithm implemented in the Python package spotpy [63] to calibrate the model automatically. Finally, the validation of the model follows a spatiotemporal strategy in which we used available discharge series: (i) in the location of the calibration stations for periods different to the calibration horizon, (ii) the locations in the drainage network different from those used for calibration, (iii) mixing strategies (i) and (ii).

### 3.3.3. Results and Discussion

We calibrated the model using the discharge information available for El Retiro stream gauging station, located in the lower part of the basin (see Figure 10C, red triangles). The period used for calibration is from January 2001 to June 2008. Figure 12A in the left column, shows the simulated discharge (blue line) overlapped with the observed data (black line). Figure 12A in the right column, presents the simulated and observed flow duration curves. The calibration results for the El Retiro station show that the estimated efficiency indexes are close to 50%, which suggests that the model reproduces the most recognizable trends of the observed data. Likewise, the flow duration curves show that the model produces optimal flows with more than a 15% probability of exceedance, including average and recession flows, covering most of the calibration time horizon. Despite the above, the model does not adequately reproduce the maximum target discharges.

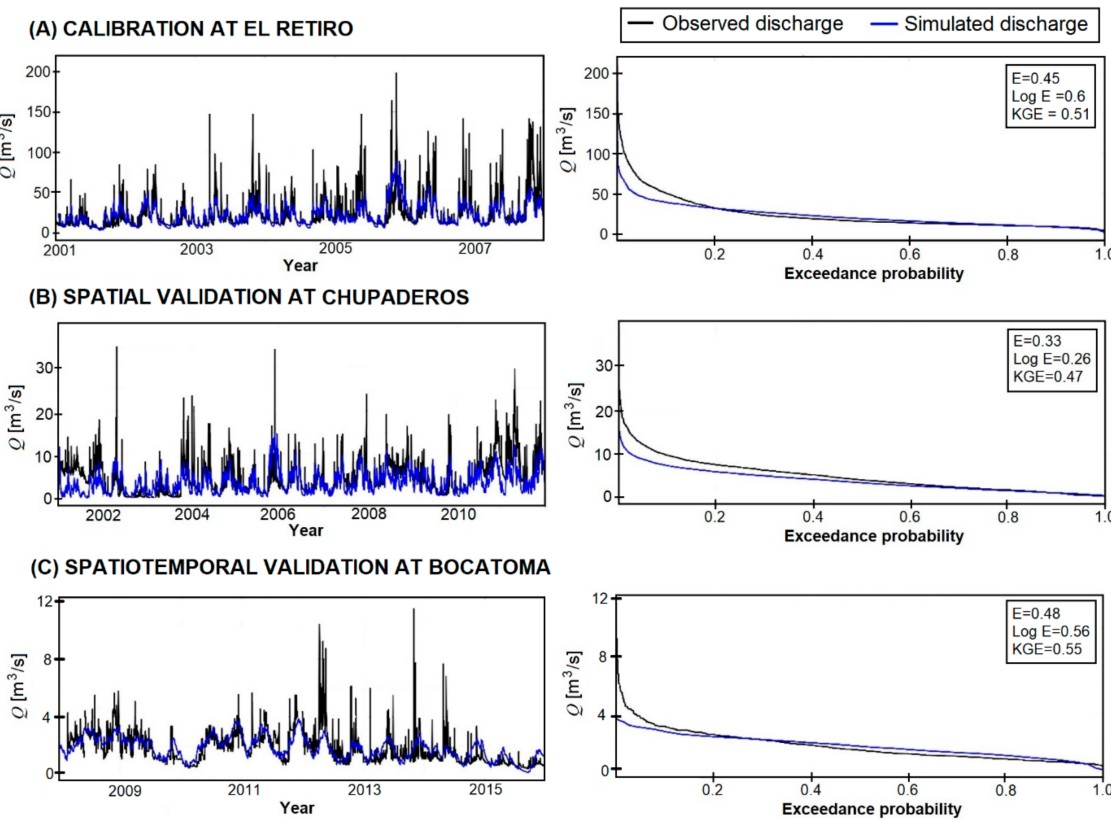

**Figure 12.** Calibration and validation results for the model implemented in WMF for the Chinchiná River basin. Row (**A**): simulated daily discharges and flow duration curve obtained in the calibration process at El Retiro station from 2001 to 2008. Row (**B**): simulated daily discharges and flow duration curve obtained in the spatial validation process at the Chupaderos station from 2001 to 2011. Row (**C**): simulated daily discharges and flow duration curve obtained in the spatial process at Bocatoma station from 2001 to 2011.

Nevertheless, this lack of representing maximum discharges is typical in continuous simulation hydrological models implemented daily due to the temporal aggregation of precipitation observed in the rain gauges. This behavior has been widely reported in the scientific literature. Thus, the calibration is acceptable from the perspective of estimating long-term changes in the driver discharge dynamics and water regulation in the basin.

With a set of calibrated parameters, we simulated discharges at the Chupaderos stream gauging station, located in the medium part of the basin upstream of Manizales (see Figure 10C). The period used for spatial validation is from June 2001 to December 2010. Figure 12, plot (B) in the left column, shows the simulated discharge (blue line) overlapped with the observed data (black line). Figure 12, plot (B) in the right column, presents the simulated and observed flow duration curves. The estimated validation efficiencies are around 33%, which indicates that the calibrated set of model parameters is spatially coherent. The simulated and observed discharge series exhibit a considerable similarity since the more noticeable trends of temporal variability are present simultaneously. This feature also shows that the rainfall estimation performed for this model is consistent since the peaks and the recession flows coincide in most of the discharge series at different locations. Its spatial distribution within the basin is representative of all the considered data sources. We obtained similar results from the spatiotemporal validation performed at the Bocatoma stream gauging station (see Figure 12C), located in the higher part of the basin upstream of the Chupaderos gauging station (see Figure 10C). In this case, the validation horizon is from June 2008 to June 2015.

### 3.3.4. Conservation Portfolios Expected Discharge Reduction

Once we calibrated and validated the model, we performed discharge simulations for the four scenarios corresponding to the proposed conservation portfolios. The objective of these simulations was to estimate discharge differences because of implementing NbS in the supplying sub-basins presented in Figures 10D and 13A. We took the trend scenario as a reference and estimated the long-term difference percentage between the simulated discharges regarding the scenarios corresponding to the conservation portfolios. Figure 13 summarizes these results. Map A presents the spatial distribution of the average discharge differences simulated for the trend scenario (without implementing NbS) regarding scenario 1. Colors from blue to red represent the estimated discharge decreasing percentage for a given stream in the supplying sub-basins. Decreases in flow, of around 25%, are expected to occur in the high Chinchiná and Rio Blanco sub-basins due to the implementation of conservation portfolios. Although not as noticeable, this discharge reduction effect is also expected in all the sub-basins included in the conservation portfolios. At the same time, plot B shows the estimated discharge decreasing percentage aggregated in the sub-basin areas, depending on the modeling scenarios constructed for each conservation portfolio. The longer the bar, the greater the discharge decreases. From these results, we conclude that implementing any of the conservation portfolios proposed by VCC produces decreases in the long-term aggregated discharges. These aggregated decreases are more noticeable in small sub-basins such as La Florest-Berlin and Los Cuervos because the NbS are implemented in the whole area. This extension in the NbS is not achievable for bigger sub-basins, so the aggregated effect masks significant discharge decreases occurring locally, as presented previously. Nevertheless, implementing conservation portfolios diminished the discharge in all cases.

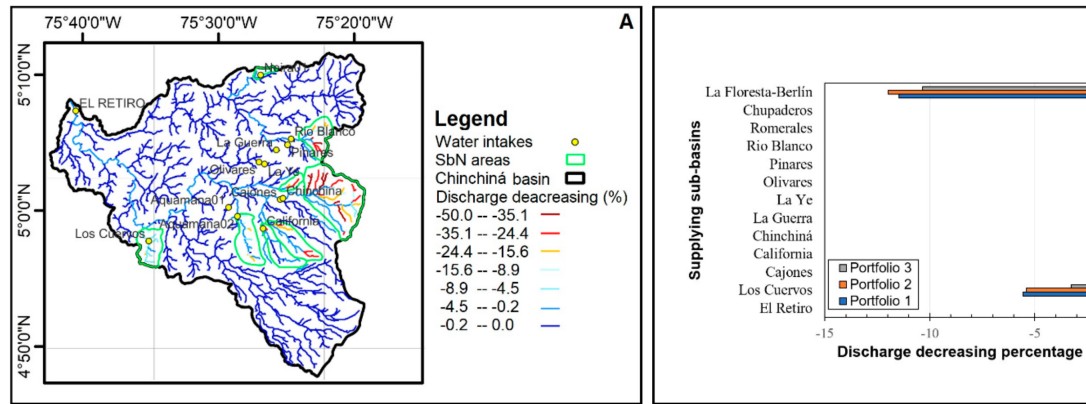

**Figure 13.** Changes in river discharges due to the implementation of NbS estimated using the TETIS models implemented for the Chinchiná river basin with the WMF utilities. Map (**A**) presents the spatial distribution of average discharge decreases simulated for trend scenarios (without implementing NbS) and scenario 1. The colors represent the estimated discharge decreasing percentage for a given stream NbS implementation areas. Plot (**B**) shows the estimated discharge decreasing percentage aggregated in the sub-basin areas, depending on the modeling scenarios constructed for each conservation portfolio.

### 3.4. QGIS Plugin and the Sediment Model

Compared to WMF, WMF-Q does not require coding skills, although it has significant limitations compared with the code version. Nevertheless, WMF-Q is an interface to WMF developed for a wider public and a faster implementation of WMF's most relevant tasks. Figure 14 shows screenshots of the plugin working with a watershed in Antioquia, Colombia.

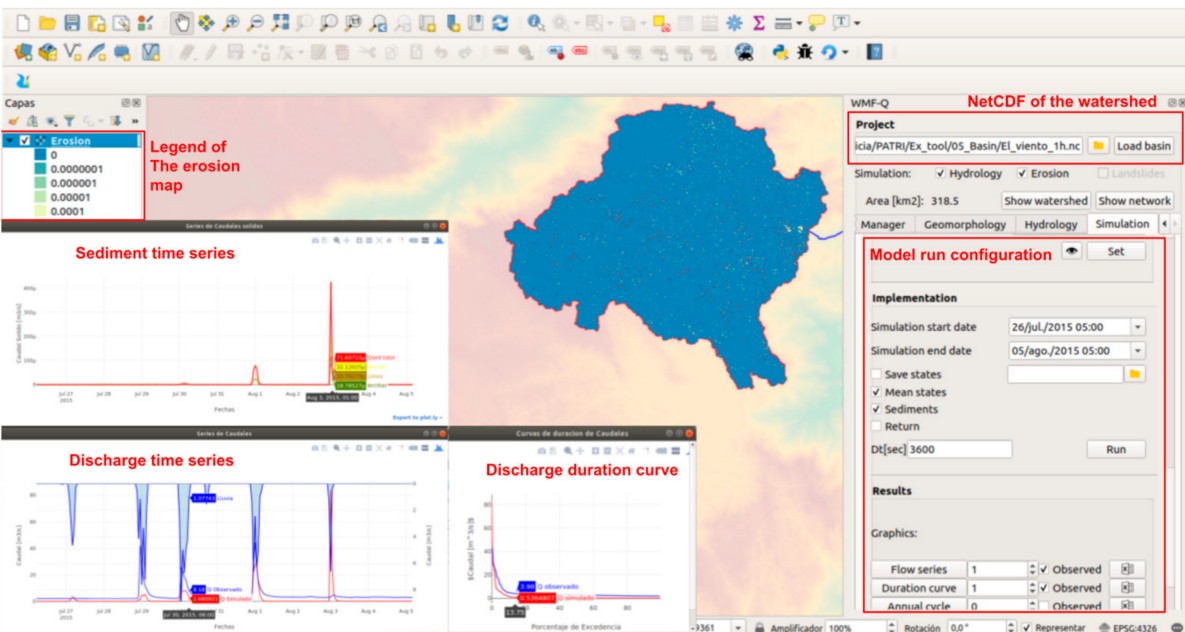

**Figure 14.** Example of WMF-Q execution simulating discharge and sediment time series. We highlight in red the elements derived from the plugin. The plugin configuration tools are on the right side. The simulated discharge and sediment time series plots are on the left. The watershed divisor and the erosion map are at the center of the image.

In this case, we ran WMF-Q for a watershed with an area of 318.5 km². Following the WMF philosophy, each project is stored in a netCDF file. In WMF-Q, the user starts by loading a previous project or delineating a new one. Once loaded, the user can edit the project in the manager tab and perform geomorphological and hydrological analysis. The simulation tab allows setting up the model parameters, configuring the rainfall inputs, and visualizing and exporting results. In the example above, we present the resulting sediment and discharge time series along with the erosion map after running the mode for three months.

In contrast with the Python interface, WMF-Q has limited capabilities. However, it offers an easier way to perform hydrological analysis and simulation. In addition, the configuration and output files keep the same format as WMF, allowing users to sketch their projects in WMF-Q and then take them to the Python interface to perform more advanced tasks. Additionally, WMF-Q allows a more straightforward introduction to hydrology. Its interaction with QGIS enables users to contrast watersheds against maps and variables, making it an ideal teaching tool.

## 4. Conclusions

This work describes WMF as a hydrological analysis and modeling tool. WMF vectorizes the watershed following its topological connectivity to execute several modeling tasks. Moreover, WMF uses common variable types such as NumPy arrays and pandas DataFrames to represent the watershed and its properties. The connection with popular Python numerical libraries and Fortran 90 at the core makes WMF a fast and expandible tool. Additionally, WMF offers a set of predefined functions that enhance its use. It also provides a model, a collection of sub-models, and functions for pre-processing from a hydrological modeling point of view. The hydrological model is a conceptual distributed model tested on different tropical and extra-tropical watersheds [32–34]. The erosion module was adapted from the work undertaken by [38] in the CASC2D-SED model. The shallow landslide corresponds to the SHIA-LANDSLIDE [35].

WMF has a wide range of modeling tools. However, we present it as a platform for developing hydrological models and analysis, taking advantage of its definitions within a scripted language. It also offers functions to link or interpolate the rainfall required for almost any modeling procedure. Additionally, WMF contains functions to store simulated states and a project's setup. In conclusion, WMF has comprehensive tools for hydrological analysis and modeling.

WMF has been used in both research and operational tasks. Operationally, WMF is used at SIATA (Sistema de Alerta Temprana de Medellin y del Valle de Aburra), an early warning system at Medellin, Colombia. WMF has also been used to research flash flood events [34], aquifer recharge analysis [64], and regional flood frequency analysis [65], and to analyze future environmental management scenarios [66]. Moreover, WMF has been used at the Universidad Nacional de Colombia and Universidad de Medellin to develop the research of several undergrad, Master's, and Ph.D. students, and extension work.

The current limitations of WMF include the lack of a parallelization scheme and the implementation of additional tools for its interaction with other software. The current hydrological model of WMF is relatively fast. However, its parallelization would allow shorter execution times in systems with multiple processors, such as High-Performance Computing (HPC) systems. Interaction through a scripted language is one of the most significant features of WMF. However, WMF is currently not linked to the Basic Model Interface (BMI) [67], which has become a standard way to interact with hydrological models in the U.S. Integrating BMI into WMF will make it more attractive to other research centers and agencies. In future developments, we expect to incorporate parallelism and BMI into WMF.

**Author Contributions:** Writing original draft preparation, N.V., writing—review and editing, O.D.Á.-V. theorical development J.I.V. and N.V., software development, N.V. and S.P.S., methodology and formal analysis, N.V. and O.D.Á.-V. funding acquisition J.I.V. and N.V. All authors have read and agreed to the published version of the manuscript.

**Funding:** The Universidad Nacional de Colombia supported the contributions of JI and SS. NV has been developing WMF since 2011 as part of his Ph.D. and SIATA work (under the grant CD511). NV continues WMF development at the Iowa Flood Center at the University of Iowa, with grants from the Iowa Department of Transportation (Grant TR-699) and the Mid-American Transportation Center (MATC).

**Data Availability Statement:** The code of WMF can be found at: https://github.com/nicolas998/WMF. The code of the examples at: https://github.com/nicolas998/WMF-Examples. Both accessed on 2 March 2023.

**Acknowledgments:** We would like to thank the reviewers and the academic editor for their insightful comments during the process of the manuscript. We also want to thank the Area Metropolitana del Valle de Aburrá who funded the SIATA project when we started building WMF.

**Conflicts of Interest:** The authors declare no conflict of interest.

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
