# Peer review of "Comprehensive Analysis of Hydrological Processes in a Programmable Environment: The Watershed Modeling Framework"

_hydrology, doi:10.3390/hydrology10040076_

Round 1

Reviewer 1 Report

See pdf

Author Response

Dear reviewer. 

Thanks for all your comments and suggestions to improve our manuscript. We have gone through each one of them. We would also like to ask for excuses for the issue related to the bibliography. It seems that during a change of format of the word document, it got messed up. We fixed it.

In the attached document, you will find our answers to your comments. 

Thanks.

Reviewer 2 Report

The overall paper is interesting and relevant. However, there are a few issues that need to be addressed before it is ready for publication.

(1) On line 42 there is a reference to the the VIC model. But the corresponding record in the references is not related to the VIC model at all. Instead there is a reference to some non peer-reviewed report in Spanish which is hard to verify and should not be included. Please add relevant references to the VIC model. e.g.: "Liang, X., Wood, E.F., & Lettenmaier, D.P. (1996). Surface soil moisture parameterization of the VIC-2L model: Evaluation and modification. Global and Planetary Change, 13(1-4), 195-206.Elsevier BV. doi: 10.1016/0921-8181(95)00046-1.". Avoid including non peer-reviewed references.

(2) On line 46-47: "Besides, models usually lack an open interface that works on a high-level programming language. " this is generally true, however it fails to acknowledge relevant work from other authors. e.g. "Salas, D., Liang, X., Navarro, M., Liang, Y., & Luna, D. (2020). An open-data open-model framework for hydrological models integration, evaluation and application. ENVIRONMENTAL MODELLING & SOFTWARE, 126, 104622.Elsevier BV. doi: 10.1016/j.envsoft.2020.104622."  More research is needed in the literature review to properly discuss previous work in the field.

(3) The references and their style is confusing as some of them are presented as a number in-between brackets which seem to be the journal style. Nevertheless, there is a significant amount of references not following that format and there are multiple instances of this mixed style. e.g. :

Lines 230: "Aristizábal (2016)", 231: "Velásquez (2020)", 238: "Salamanca, (2020)", 251: "Julien (1998)", 259: "Velásquez et al. (2020)". The reference style must follows the journal convention consistently throughout  the paper.

(4) On line 299: "The function also computes the travel time using eight different equations", Which equations were used?

(5) There is a typo in line 573 "Temrpana" instead of "Temprana".

(6) There is an extra dot at the end of line 598.

(7) The acknowledgments section seems to be the instructions from the journal template.

(8) Line 602 there is an extra closing quote sign at the end.  

Author Response

Dear Reviewer. 

We are thankful for your comments on our manuscript. We took care of each one of them. Also, we would like to ask for excuses regarding the bibliography. It seems that it got messed up when we changed the format of the word document. We fixed it. 

In the attached document, you will find our answers to your comments. 

Thanks.
